# Multi-Marginal Schrödinger Bridge Matching

## Abstract

Understanding the continuous evolution of populations from discrete temporal snapshots is a critical research challenge, particularly in fields like developmental biology and systems medicine where longitudinal tracking of individual entities is often impossible. Such trajectory inference is vital for unraveling the mechanisms of dynamic processes. While Schrödinger Bridge (SB) offer a potent framework, their traditional application to pairwise time points can be insufficient for systems defined by multiple intermediate snapshots. This paper introduces Multi-Marginal Schrödinger Bridge Matching (MSBM), a novel algorithm specifically designed for the multi-marginal SB problem. MSBM extends iterative Markovian fitting (IMF) to effectively handle multiple marginal constraints. This technique ensures robust enforcement of all intermediate marginals while preserving the continuity of the learned global dynamics across the entire trajectory. Empirical validations on synthetic data and real-world single-cell RNA sequencing datasets demonstrate the competitive or superior performance of MSBM in capturing complex trajectories and respecting intermediate distributions, all with notable computational efficiency.

## 1 Introduction

Understanding the continuous evolution of populations from discrete temporal snapshots represents a significant challenge in various scientific disciplines, particularly in fields like developmental biology [7, 42] and systems medicine [29] where tracking individual entities longitudinally is often unfeasible. The ability to infer trajectories from such snapshot data is crucial for elucidating the underlying mechanisms of dynamic processes. The Schrödinger Bridge (SB) problem, originally rooted in statistical mechanics [43], has garnered substantial interest in machine learning as an entropy-regularized, continuous-time formulation of optimal transport [20, 30]. It seeks to identify the most probable evolutionary path between prescribed initial and terminal distributions, and has been successfully employed in generative modeling [3, 4, 9, 26, 27, 37, 38, 45, 49].

However, many real-world scenarios present observations or constraints at multiple time points, not just at the beginning and end of a process. For instance, in single-cell RNA sequencing (scRNA-seq) experiments, which are pivotal for studying complex biological processes like cell differentiation, cells are typically destroyed upon measurement [6, 17, 28]. This destructive nature makes it impossible to track individual cells over time, thus necessitating the inference of developmental trajectories from population-level snapshots collected at several intermediate stages. Similarly, meteorological systems may have partial observations across various times [11, 32]. Such situations necessitate a multi-marginal generalization of the SB problem (mSBP), where the path measure must align with prescribed marginal distributions at multiple intermediate time points. While the traditional SB framework offers a powerful approach, its standard application to pairwise time points can prove insufficient for systems characterized by multiple intermediate snapshots. Although more specialized methods for mSBP have recently been developed [8, 18, 44], the direct application of some multi-marginal approaches can lead to error accumulation if not carefully managed, particularly

when learned controls are even slightly inaccurate. These challenges highlight the need for robust and scalable solutions for the mSBP that can effectively integrate information across all observed time points.

This paper introduces Multi-Marginal Schrödinger Bridge Matching (MSBM), a novel algorithm specifically developed to address the multi-marginal SB problem by building upon and extending the Iterative Markovian Fitting (IMF) algoritmhs [36, 45]. MSBM is designed to effectively manage multiple marginal constraints by constructing local SBs on each interval and seamlessly integrating them. This local construction strategy, underpinned by a shared global parametrization of control functions, ensures the robust enforcement of all intermediate marginal distributions while crucially preserving the continuity of the learned global dynamics across the entire trajectory. Empirical validations conducted on synthetic datasets as well as real-world single-cell RNA sequencing data demonstrate that MSBM achieves competitive or superior performance in capturing complex trajectories and accurately respecting intermediate distributions, all while exhibiting notable computational efficiency. Our work aims to provide a robust and scalable computational method for these multi-marginal settings, addressing the critical need for consistent and tractable dynamic inference when data is available as snapshots at multiple time points.

We summarize our contributions as follows:

- We extend the theoretical and algorithmic foundations of SBs, including the IMF iteration and optimal control perspectives, to the challenging multi-marginal setting.

- We introduce an efficient modeling approach for trajectory inference, that constructs and smoothly integrates local SBs across sub-intervals, inherently allows for parallelized training, leading to significant speed-ups.

- Through comprehensive experiments on both synthetic and real-world single-cell RNA sequencing data, we demonstrate that MSBM accurately models complex population dynamics and outperforms state-of-the-art methods in both trajectory fidelity and computational speed.

**Notation.** Let $\mathcal{P}_{[0,T]}$ denote the space of continuous functions taking values in $\mathbb{R}^d$ on the interval $[0,T]$. We use an uppercase letter $\mathbb{P} \in \mathcal{P}_{[0,T]}$ to represent a path measure. For a path measure $\mathbb{P} \in \mathcal{P}_{[0,T]}$, the marginal distribution at discrete time points $\mathcal{T} = \{t_0, \ldots, t_k\}$, where $0 = t_0 < t_1 < \cdots < t_k = T$ is denoted by $\mathbb{P}_\mathcal{T} \in \mathcal{P}_\mathcal{T}$, where we define $\mathcal{P}_\mathcal{T}$ as the set of measures $\mathbb{P}$ over $\mathbb{R}^{d \times |\mathcal{T}|}$. Additionally, the conditional distribution of $\mathbb{P}$, given $\mathcal{T}$, is denoted by $\mathbb{P}_{|\mathcal{T}} \in \mathcal{P}_{[0,T]}$. Moreover, a path measure $\mathbb{P}$ can be defined as mixture. For any Borel measurable set $A \in \mathcal{B}(\Omega)$, $\mathbb{P}$ can be defined by $\mathbb{P}(A) = \int_{\mathbb{R}^{d \times |\mathcal{T}|}} \mathbb{P}_{|\mathcal{T}}(A|\mathbf{x}_\mathcal{T}) d\mathbb{P}_\mathcal{T}(\mathbf{x}_\mathcal{T})$, where $\mathbb{P} \in \mathcal{P}_{0,T}$ and $\mathbb{P} \in \mathcal{P}_\mathcal{T}$, and we use the shorthand $\mathbf{x}_\mathcal{T} := (\mathbf{x}_1, \cdots, \mathbf{x}_k)$ and $[0:k] := \{0, 1, \cdots, k\}$. The Kullback-Leibler (KL) divergence between two probability measures $\mu$ and $\nu$ on space $\mathcal{X}$ is defined as $\mathrm{D}_{\mathrm{KL}}(\mu|\nu) = \int_\mathcal{X} \log \frac{d\mu}{d\nu}(\mathbf{X}) d\mu(\mathbf{X})$ when $\mu$ is absolutely continuous with respect to $\nu$ ($\mu \ll \nu$), and $\mathrm{D}_{\mathrm{KL}}(\mu|\nu) = +\infty$ otherwise. We will often refer to probability measures on $\mathbb{R}^d$ and their Lebesgue densities interchangeably, under the standard assumption of absolute continuity. Finally, for a function $\mathcal{V} : [0,T] \times \mathbb{R}^d \to \mathbb{R}$, we define the gradient and laplcaian operators with respect to $\mathbf{x} \in \mathbb{R}^d$ as $\nabla \mathcal{V}$ and $\Delta \mathcal{V}$, respectively, and its partial derivative with respect to time $t \in [0,T]$ as $\partial_t \mathcal{V}$.

## 2 Preliminaries

### 2.1 Schrödinger Bridge Matching (SBM)

The Schrödinger Bridge problem (SBP) [16, 43] is a stochastic optimal transport problem [30] that seeks the optimal transport plan for endpoint marginals $\rho_0$ and $\rho_T$. In this paper, we focus on the dynamical representation, where a reference distribution $\mathbb{Q} \in \mathcal{P}_{[0,T]}$ is induced by the SDEs:

$$d\mathbf{X}_t = f_t(\mathbf{X}_t)\,dt + \sigma\,d\mathbf{W}_t, \quad \mathbf{X}_0 \sim \rho_0, \tag{1}$$

where $f_t : \mathbb{R}^d \to \mathbb{R}^d$ is a drift, $\sigma \in \mathbb{R}$ is a diffusion, and $\mathbf{W}_t \in \mathbb{R}^d$ is a standard Wiener process. With the base reference path measure $\mathbb{Q}$, the dynamic representation of the SB [20, 35, 39] is:

$$\min_{\mathbb{P} \in \mathcal{P}_{[0,T]}} \mathrm{D}_{\mathrm{KL}}(\mathbb{P}|\mathbb{Q}), \quad \text{subject to} \quad \mathbb{P}_0 \sim \rho_0, \quad \mathbb{P}_T \sim \rho_T. \tag{SBP}$$

Recent advancements in dynamical optimal transport [37, 45] have introduced a novel numerical methodology for solving SBP. This approach reframes SBP by decomposing its dynamical constraints into the time-evolving marginal distributions $\mathbb{P}_t$ for all $t \in [0, T]$ and the joint coupling $\mathbb{P}_{0,T}$. This optimization relies on IMF [45], a technique that iteratively refines the path measure $\mathbb{P} \in \mathcal{P}_{[0,T]}$. IMF alternates between two projection called Markovian and Reciprocal projections to preserve the correct endpoint marginals $(\rho_0, \rho_T)$ throughout the optimization.

**Reciprocal Projection $\mathcal{R}$.** For a given reference measure $\mathbb{Q}$ from (1), and a path measure $\mathbb{P}$ with marginals specified at end points $\mathcal{T} = \{0, T\}$ the reciprocal projection is defined as:

$$\mathcal{R}(\mathbb{P}, \mathcal{T}) := \mathbb{P}_{\mathcal{T}}\mathbb{Q}_{|\mathcal{T}} = \mathbb{P}_{0,T}\mathbb{Q}_{|0,T}. \tag{2}$$

This projection constructs a new path measure by taking the endpoint coupling $\mathbb{P}_{0,T}$ from $\mathbb{P}$ and forming a mixture of bridge process using $\mathbb{Q}$ conditioned on these end points. Sampling from $\Pi := \mathcal{R}(\mathbb{P}, \mathcal{T})$ involves drawing end points samples $(\mathbf{X}_0, \mathbf{X}_T) \sim \mathbb{P}_{0,T}$ and then generating a path $\mathbf{X}_t^{\mathcal{T}}$ between them using conditional reference measure $\mathbb{Q}_{|0,T}$ which induced by following SDEs, for any $(\mathbf{x}_0, \mathbf{x}_T)$:

$$d\mathbf{X}_t^{\mathcal{T}} = \left[ f_t(\mathbf{X}_t^{\mathcal{T}}) + \sigma^2 \nabla \log \mathbb{Q}_{T|t}(\mathbf{x}_T | \mathbf{X}_t^{\mathcal{T}}) \right] dt + \sigma d\mathbf{W}_t, \quad \mathbf{X}_0^{\mathcal{T}} = \mathbf{x}_0, \tag{3}$$

If $\mathbb{Q}_{|0,T}$ has tractable bridge formulation, for example, when $\mathbb{Q}$ is chosen as a Brownian motion *i.e.*, $d\mathbf{X}_t = \sigma d\mathbf{W}_t$, sampling the path at time $t$ given the endpoints can be performed as:

$$\mathbf{X}_t^{\mathcal{T}} \sim \mathcal{N}\left( (1 - \tfrac{t}{T})\mathbf{X}_0 + \tfrac{t}{T}\mathbf{X}_T, t(1 - \tfrac{t}{T})\sigma^2 \right), \quad \text{where } (\mathbf{X}_0, \mathbf{X}_T) \sim \mathbb{P}_{0,T}. \tag{4}$$

**Markov Projection $\mathcal{M}$.** Although the reciprocal projection $\mathcal{R}$ in (2) preserves end point marginals $(\rho_0, \rho_T)$, its sampling process in (4) requires both $(\mathbf{X}_0, \mathbf{X}_T)$, making it non-Markovian and thus ill-suited for generative modeling aimed at sampling from $\rho_T$ without knowing $\mathbf{X}_T$. The Markov projection $\mathcal{M}$ resolves this by projecting $\Pi := \mathcal{R}(\mathbb{P}, \mathcal{T})$ into a family of Markov process while ensuring $\mathbb{P}^\star = \Pi_t$ for all $t \in [0, T]$. Again, when $\mathbb{Q}$ is chosen as a Brownian motion *i.e.*, $d\mathbf{X}_t = \sigma d\mathbf{W}_t$, the Markov projection of $\Pi$, $\mathbb{P}^\star = \mathcal{M}(\Pi, \mathcal{T})$, is induced by following SDEs:

$$d\mathbf{X}_t^\star = \sigma v^\star(t, \mathbf{X}_t^\star) dt + \sigma d\mathbf{W}_t, \quad \mathbf{X}_0^\star \sim \Pi_0, \tag{5}$$

$$\text{where} \quad v^\star(t, \mathbf{x}) = \tfrac{1}{T-t} \left( \mathbb{E}_{\mathbb{Q}_{T|t}}\left[ \mathbf{X}_T | \mathbf{X}_t = \mathbf{x} \right] - \mathbf{x} \right). \tag{6}$$

Intuitively, the term $\mathbb{E}_{\mathbb{Q}_{T|t}}\left[ \mathbf{X}_T | \mathbf{X}_t = \mathbf{x} \right]$ can be understood as a prediction of the target state $\mathbf{X}_t^\star$. Flow matching [23] of Bridge matching [37] tackles the approximation $\mathbf{X}_T^\star \approx \mathbb{E}_{\mathbb{Q}_{T|t}}\left[ \mathbf{X}_T | \mathbf{X}_t = \mathbf{x} \right]$ by learning a drift function. This learned drift guides the evolution of $\mathbf{X}_t^\star$ such that its terminal state aligns with the target, often by regressing the drift agains a target drift derived from samples of $(\mathbf{X}_0, \mathbf{X}_T)$ under the reference conditional bridge measure $\mathbb{Q}_{|0,T}$.

Building upon the projections $\mathcal{R}$ and $\mathcal{M}$, Schrödinger Bridge Matching (SBM) methods [37, 45] refines the path measure through an alternating iteraive procedure:

$$\mathbb{P}^{(2n+1)} := \mathcal{M}(\mathbb{P}^{(2n)}, \mathcal{T}), \ \mathbb{P}^{(2n+2)} := \mathcal{R}(\mathbb{P}^{(2n+1)}, \mathcal{T}). \tag{7}$$

Initialized with $\mathbb{P}^{(0)} = \mathbb{P}_{\mathcal{T}}^{(0)} \mathbb{Q}_{|0,T}$, utilizing $\mathbb{P}_{\mathcal{T}}^{(0)}$ is independent coupling of $\rho_0$ and $\rho_T$ along with the reference conditional bridge measure $\mathbb{Q}_{|\mathcal{T}}$. Please refer to [37, 45] for more details.

# 3 Multi-Marginal Iterative Markovian Fitting

Dynamic SB methods, as discussed in Section 2, have traditionally focused on problems defined by two endpoint marginal distributions, $(\rho_0, \rho_T)$. However, in real-world applications, particularly in fields like developmental biology (e.g., scRNA-seq studies of cellular differentiation), systems are often observed through snapshots at multiple intermediate time points, not just at the beginning and end of a process. This prevalence of multi-stage data highlights a critical limitation of standard SB approaches. While the theoretical extension of SB methods to handle multiple marginals has been explored [1, 31], the development of robust and scalable computational methods for these multi-marginal settings has lagged. Recently, methods with IPF-type objectives have been derived for multi-marginal cases [8, 44]. However, challenges persist in ensuring global dynamic consistency across all intervals, maintaining computational tractability as the number of marginals increases.

In this section, we extends the SBM framework−conventionally applied to problems with two endpoint marginals $(\rho_0, \rho_T)$ and foundational to IMF methods−to handle cases involving $k+1$ multiple snapshots $(\rho_0, \rho_{t_1}, \cdots, \rho_T)$ on discrete time stamps $\mathcal{T} = \{t_0, t_1, \cdots, t_k\}$ where $0 = t_0 < t_1 < \cdots < t_k = T$[1]. Similar to SBP, the dynamic multi-marginal Schrödinger Bridge problem can be formally defined as [10] the entropy minimization problem:

$$\min_{\mathbb{P} \in \mathcal{P}_{[0,T]}} \mathrm{D}_{KL}(\mathbb{P}|\mathbb{Q}), \quad \text{subject to} \quad \mathbb{P}_t \sim \rho_t, \quad \forall t \in \mathcal{T}. \quad \text{(mSBP)}$$

To find a most probable path $\mathbb{P}^{\mathrm{mSBP}}$, the solution of mSBP under multiple constraints, we will generalize the principles of SBM in Section 2.1 to the multi-marginal cases in Section 3.1. The extension of dynamic SB optimality [20, 35] and the associated stochastic optimal control problem [39] to multi-marginal settings is presented in Appendix A.

## 3.1 Multi-Marginal Projection operators

To develop multi-marginal extension of SBM, we investigate how the IMF framework can be adapted to scenarios with multiple snapshots (*i.e.*, where the set of time points $\mathcal{T}$ has cardinality $|\mathcal{T}| > 2$). This adaptation necessitates extending the fundamental building blocks of SBM—specifically, the reciprocal projection $\mathcal{R}$ and the Markov projection $\mathcal{M}$—to handle multiple marginal constraints.

**Multi-Marginal Reciprocal Projection** $\mathcal{R}^{\mathrm{mm}}$. First, we state and prove a proposition that character-izes the reciprocal structure of conditional path measures. In particular, we focus on a mixture of bridges $\Pi = \Pi_{\mathcal{T}} \mathbb{Q}_{|\mathcal{T}} \in \mathbb{P}_{[0,T]}$ constrained by the marginals at multiple timestamps in $\mathcal{T}$.

**Proposition 1** (Reciprocal Property). *For any* $\mathbf{x}_{\mathcal{T}} := (\mathbf{x}_0, \mathbf{x}_{t_1}, \cdots, \mathbf{x}_T) \in \mathbb{R}^{d \times (k+1)}$ *and* $t \in [t_{i-1}, t_i)$, *the marginal distribution of* $\mathbb{Q}_{|\mathcal{T}}(\cdot|\mathbf{x}_{\mathcal{T}})$ *at* $t$ *satisfies:*

$$\mathbb{Q}_{|\mathcal{T}}(\mathbf{x}_t|\mathbf{x}_{\mathcal{T}}) = \mathbb{Q}_{|t_{i-1}, t_i}(\mathbf{x}_t|\mathbf{x}_{t_i}, \mathbf{x}_{t_{i-1}}). \quad (8)$$

*Therefore, for any* $\mathbb{P} \in \mathcal{P}_{[0,T]}$ *the reciprocal projection* $\mathcal{R}^{\mathrm{mm}}(\mathbb{P}, \mathcal{T})$ *admits the following factorization:*

$$\mathcal{R}^{\mathrm{mm}}(\mathbb{P}, \mathcal{T}) = \mathbb{P}_{\mathcal{T}}\mathbb{Q}_{|\mathcal{T}} = \mathbb{P}_{t_0, \cdots, t_k}\mathbb{Q}_{|t_0, \cdots, t_k} = \mathbb{P}_{t_0, \cdots, t_k} \prod_{i=1}^{k} \mathbb{Q}_{|t_{i-1}, t_i}, \quad \mathbb{P}\text{-a.e.} \quad (9)$$

A key implication of the reciprocal property, detailed in Proposition 1, is that a mixture of diffusion bridges constrained on $\mathcal{T}$ factorizes into independent segments over successive time intervals. This factorization simplifies the analysis and simulation of the overall path measure. Since each segment can then be treated as a standard conditional bridge process as in (3), closed-form sampling, such as in (4), can be applied independently in parallel to each subinterval $\{t_{i-1}, t_i\}_{i \in [1:k]}$. This tractability is essential for developing an efficient multi-marginal SBM algorithm.

**Multi-Marginal Markov Projection** $\mathcal{M}^{\mathrm{mm}}$. With the reciprocal property and factorization in (9), we show that the Markov projection on multi-marginal case can be constructed by similar fashion.

**Proposition 2** (Multi-Marginal Markovian Projection). *Let* $\Pi \in \mathcal{P}_{[0,T]}$ *admit factorzation in* (9). *The multi-marginal Markov projection of* $\Pi$, $\mathbb{P}^{\star} := \mathcal{M}^{\mathrm{mm}}(\Pi, \mathcal{T}) \in \mathcal{P}_{[0,T]}$, *is associated with the SDE:*

$$d\mathbf{X}_t^{\star} = [f_t(\mathbf{X}_t^{\star}) + \sigma v^{\star}(t, \mathbf{X}_t^{\star})] dt + \sigma d\mathbf{W}_t, \quad \mathbf{X}_0^{\star} \sim \Pi_0, \quad (10)$$

$$\text{where } v^{\star}(t, \mathbf{x}) = \sum_{i=1}^{k} \mathbf{1}_{[t_{i-1}, t_i)} \mathbb{E}_{\Pi_{t_i|t}} \left[ \nabla \log \mathbb{Q}_{t_i|t}(\mathbf{X}_{t_i}|\mathbf{X}_t)|\mathbf{X}_t = \mathbf{x} \right]. \quad (11)$$

*Moreover,* $v^{\star}$ *satisfies the Fokker-Planck equation (FPE) [40]:*

$$\partial_t \rho_t = -\nabla \cdot (v_t^{\star}(\mathbf{x})\rho_t(\mathbf{x})) + \frac{\sigma^2}{2}\Delta \rho_t(\mathbf{x}) = 0, \quad \rho_t = \Pi_t, \quad \forall t \in \mathcal{T}, \quad (12)$$

*where* $p_t$ *is marginal density of* $\Pi_t$. *In other words,* $\mathbb{P}_t^{\star} = \Pi_t$ *for all* $t \in [0, T]$. *d*

As established in Proposition 2, constructing a global diffusion process via (10) with the optimal control $v^{\star}$ (11)) yields a multi-marginal Markov projection $\mathbf{X}_{[0,T]}^{\star}$ that is continuous over the entire time interval $[0, T]$. The continuity arises because the local Markov projections, $\mathbf{X}_{[t_{i-1}, t_i]}^{\star}$, on each sub-interval are derived from factorized conditional bridge $\mathbb{Q}_{|t_{i-1}, t_i}$ in (9). These bridges are

---

[1]Our framework accommodates arbitrary time intervals between successive time stamps.

anchored by identical marginal distributions at there shared boundaries; for instance, both $\mathbf{X}^{\star}_{[t_{i-1}, t_i]}$ and $\mathbf{X}^{\star}_{[t_i, t_{i+1}]}$ is guaranteed to match the marginal distribution $\rho_{t_i}$ at time $t_i$. Consequently, these local diffusion processes connect seamlessly at adjacent timestamps, resulting in a smooth and well-defined path for $\mathbf{X}^{\star}_{[0,T]}$. The well-defined nature of the global path, in conjunction with the projections $\mathcal{R}^{\mathrm{mm}}$ and $\mathcal{M}^{\mathrm{mm}}$, is fundamental to successfully applying the SBM framework to the mSBP. Finally, the uniqueness condition for standard SB [45, Proposition 5] can also be extended to multi-marginal case.

**Proposition 3** (Uniqueness). *Let $\mathbb{P}^{\star}$ be a Markov measure which is reciprocal class of $\mathbb{Q}$ satisfying $\mathbb{P}^{\star}_t = \rho_t$ for all $t \in \mathcal{T}$. Then, $\mathbb{P}^{\star}$ is unique solution $\mathbb{P}^{mSBP}$ of the mSBP.*

Building on the projection operators $\mathcal{R}^{\mathrm{mm}}, \mathcal{M}^{\mathrm{mm}}$ with the uniqueness result of Proposition 3, we can apply the iterative algorithm used in SBM algorithm [45, Algorithm 1] to the multi-marginal setting:

$$\mathbb{P}^{(2n+1)} := \mathcal{M}^{\mathrm{mm}}(\mathbb{P}^{(2n)}, \mathcal{T}), \ \mathbb{P}^{(2n+2)} := \mathcal{R}^{\mathrm{mm}}(\mathbb{P}^{(2n+1)}, \mathcal{T}), \quad |\mathcal{T}| > 2. \tag{13}$$

The convergence guarantees proved for the iteration apply equally well to the multi-marginal case.

**Proposition 4** (Convergence). $\mathbb{P}^{(n)} = \mathbb{P}^{mSBP}$ *of mSBP as $n \uparrow \infty$ with iterative procedure in* (13).

### 3.2 Practical Implementation.

In practice, at each iteration $n$ of (13) we approximate the optimal control $v^{\star}$ from (11) by a neural network $v_{\theta}$. By Girsanov theorem, $\theta$ are chosen to minimize the following training objective function:

$$\mathcal{L}(\theta, \mathcal{T}, \Pi_{\mathcal{T}}) = \int_0^T \mathbb{E}_{\Pi_{t,\mathcal{T}}}[||\sigma \nabla \log \mathbb{Q}_{\beta_{\mathcal{T}}(t)|t}(\mathbf{X}_{\beta_{\mathcal{T}}(t)}|\mathbf{X}_t) - v_{\theta}(t, \mathbf{X}_t)||^2 dt], \tag{14}$$

where $\beta_{\mathcal{T}}(t) = \min_u\{u > t | t \in \mathcal{T}\} \in [0, T]$ is the most recent time point in $\mathcal{T}$ after time $t$. With this notation, the SBM can be generalized to the case of multi-marginal constraints. For example, when $\mathcal{T} = \{0, T\}$ then (14) reduces to the objective function described in [45].

The learned Markov control $v_{\theta^{\star}}(t, \mathbf{x}_t)$ then ensures $\mathbb{P}^{\theta^{\star}}_t = \Pi_t$ for all $t \in [0, T]$. Moreover, prior SBM algorithms interleave forward and backward-time Markov projections to re-anchor the terminal distribution and prevent bias between $\mathbb{P}^{(n)}_T$ and $\Pi_T$ accumulate for each $n \in \mathbb{N}$. In the multi-marginal setting, we again build the backward-time Markov projection as in Proposition 2 by *gluing* the local bridge reversals, so that $\mathbb{P}^{\star}$ is governed by both SDEs (10) and the corresponding backward dynamics:

$$d\mathbf{Y}^{\star}_t = [-f_{T-t}(\mathbf{Y}^{\star}_t) + \sigma u^{\star}(t, \mathbf{Y}^{\star}_t)] dt + \sigma d\mathbf{W}_t, \quad \mathbf{Y}^{\star}_0 \sim \Pi_T, \tag{15}$$

$$\text{where } u^{\star}(t, \mathbf{y}) = \sum_{i=1}^k \mathbf{1}_{(t_{i-1}, t_i]}(t) \mathbb{E}_{\Pi_{t|t_{i-1}}}\left[\nabla \log \mathbb{Q}_{t|t_{i-1}}(\mathbf{Y}_t|\mathbf{Y}_{t_{i-1}})|\mathbf{Y}_t = \mathbf{y}\right], \tag{16}$$

where the backward optimal control $u^{\star}$ in (16) can be approximated with neural network $u_{\phi}$ where $\phi$ is chosen to minimize the following training objective function with $\gamma_{\mathcal{T}}(t) = \max_u\{u < t | t \in \mathcal{T}\}$:

$$\mathcal{L}(\phi, \mathcal{T}, \Pi_{\mathcal{T}}) = \int_0^T \mathbb{E}_{\Pi_{t,\mathcal{T}}}[||\sigma \nabla \log \mathbb{Q}_{t|\gamma_{\mathcal{T}}(t)}(\mathbf{Y}_t|\mathbf{Y}_{\gamma_{\mathcal{T}}(t)}) - u_{\phi}(t, \mathbf{Y}_t)||^2 dt]. \tag{17}$$

## 4 Multi-Marginal Schrödinger Bridge Matching

A naïve extension of the standard SBM using, multi-marginal projections $\mathcal{R}^{\mathrm{mm}}$ and $\mathcal{M}^{\mathrm{mm}}$ in Sec 3, encounters significant limitations not present in the traditional two-endpoint setting. In such an extension, each iteration typically enforces marginal constraints only at the global endpoints $(\rho_0, \rho_T)$. The multi-marginal coupling $\Pi^{(n)}_{\mathcal{T}}$ at each iteration $n$ of (13) is then derived by propagating the projected dynamics in (10) or (15) solely from these end points $\rho_0$ or $\rho_T$, respectively.

This approach leads to critical issues specific to the multi-marginal context. Firstly, if the learned controls, such as $v^{\star}$ (forward) or $u^{\star}$ (backward), are even slightly inaccurate, significant biases can arise between the inferred intermediate marginals $(\Pi^{(n)}_{t_1}, \cdots \Pi^{(n)}_{t_{k-1}})$ and the target marginals $(\rho_{t_1}, \cdots, \rho_{t_{k-1}})$. Secondly, these discrepancies tend to accumulate iteratively. This accumulation is exacerbated because, beyond an initialization $\Pi^{(0)} = \mathbb{P}^{(0)}_{\mathcal{T}} \mathbb{Q}_{|\mathcal{T}}$ with $\mathbb{P}^{(0)}_{\mathcal{T}}$, independent joint coupling of $\{\rho_t\}_{t \in \mathcal{T}}$, where the joint distribution might be informed by all prescribed data distributions, the subsequent self-refinement process for the dynamics often does not directly incorporate the

**Algorithm 1** Training of MSBM

1: **Input:** Snapshots $\{\rho_t\}_{t\in\mathcal{T}}$, bridge $\mathbb{Q}_{|\mathcal{T}}$, $N \in \mathbb{N}$
2: Let $\{\mathbb{P}^{(0)}_{\mathcal{T}_i}\}_{i\in[1:k]}$ joint coupling of $\{\rho_{t\in\mathcal{T}_i}\}_{i\in[1:k]}$.
3: **for** $n \in \{0, \dots, N-1\}$ **do**
4:     **for** $i \in \{1, \dots, k-1\}$ **do in parallel**
5:         Let $\Pi^{(2n)}_{\mathcal{T}_i} = \mathbb{P}^{(2n)}_{\mathcal{T}_i}$
6:         Estimate $\mathcal{L}(\phi, \mathcal{T}_i, \Pi^{(2n)}_{\mathcal{T}_i}, \mathbb{Q}_{|\mathcal{T}_i})$
7:         Estimate $\tilde{\mathcal{L}}(\phi) = \sum_{i=1}^k \mathcal{L}(\phi, \mathcal{T}_i, \Pi^{(2n)}_{\mathcal{T}_i}, \mathbb{Q}_{|\mathcal{T}_i})$
8:         $u_{\phi^\star} = \arg\min_\phi \sum_{i=1}^k \tilde{\mathcal{L}}(\phi)$
9:         Simulate local backward SBs $\{\mathbb{P}^{i,(2n+1)}\}_{i\in[1:k]}$
10:     **for** $i \in \{1, \dots, k-1\}$ **do in parallel**
11:         Let $\Pi^{(2n+1)}_{\mathcal{T}_i} = \mathbb{P}^{(2n+1)}_{\mathcal{T}_i}$
12:         Estimate $\mathcal{L}(\theta, \mathcal{T}_i, \Pi^{(2n+1)}_{\mathcal{T}_i}, \mathbb{Q}_{|\mathcal{T}_i})$
13:         Estimate $\tilde{\mathcal{L}}(\theta) = \sum_{i=1}^k \mathcal{L}(\theta, \mathcal{T}_i, \Pi^{(2n+1)}_{\mathcal{T}_i}, \mathbb{Q}_{|\mathcal{T}_i})$
14:         $v_{\theta^\star} = \arg\min_\theta \sum_{i=1}^k \mathcal{L}(\theta, \mathcal{T}_i, \Pi^{(2n+1)}_{\mathcal{T}_i})$
15:         Simulate local forward SBs $\{\mathbb{P}^{i,(2n+2)}_{[t_{i-1}, t_i]}\}$
16: **end for**
17: **Output:** $v_\theta^\star$, $u_\phi^\star$

**Algorithm 2** Simulation of MSBM (forward)

**Input:** Initial $\rho_0$, learned control $v_{\theta^\star}$
Sample $\mathbf{X}_0 \sim \rho_0$
Simulate forward SDE over $[0, T]$
$d\mathbf{X}_t^\star = [f_t + \sigma v_{\theta^\star}(t, \mathbf{X}_t^\star)]\, dt + \sigma d\mathbf{W}_t,$
**Output:** Trajectory $\mathbf{X}_{[0,T]}^\star$

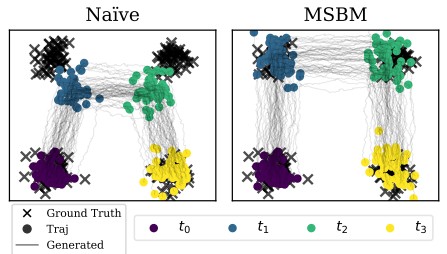

Figure 1: **(Left)** The naïve extension fails to model intermediate states due to the accumulation of errors. **(Right)** In contrast, MSBM successfully models the ground truth data.

intermediate data distributions $(\rho_{t_1}, \cdots, \rho_{t_{k-1}})$ into its training objective except $\rho_0$ and $\rho_T$. Without explicit targets for the intermediate marginals guiding each iteration, the inferred paths between $\rho_0$ and $\rho_T$ can "collapse" or drift away from the desired states. Consequently, precisely satisfying all intermediate constraints becomes increasingly challenging as iterations proceed.

To address this issue of error accumulation and ensure all marginal constraints $\{\rho_t\}_{t\in\mathcal{T}}$ are satisfied, we propose a method that involves constructing local SBs on each interval $[t_{i-1}, t_i]$ and then seamlessly *gluing* them together. Instead of propagating dynamics from the global endpoints $\rho_0$ and $\rho_T$ alone, our approach first establishes local SBs for each segment. The resulting local couplings are then systematically integrated to satisfy all specified marginal distributions $\{\rho_t\}_{t\in\mathcal{T}}$ across the entire time interval $[0, T]$. This local construction strategy helps prevent the compounding of errors at intermediate time points while still aiming to achieve the overall multi-marginal SB solution, $\mathbb{P}^{\text{mSBP}}$. The theoretical basis is provided by the following result.

**Corollary 5** (Multi-Marginal Schrödinger Bridge). *Assume a sequence of controls $\{v^i, u^i\}_{i\in[1:k]}$, where each $v^i, u^i$ induced local SBs $\mathbb{P}^i$ of SBP over local interval $[t_{i-1}, t_i]$ with distributions $(\rho_{t_{i-1}}, \rho_{t_i})$ in a forward and backward direction, respectively. If $\lim_{t\uparrow t_i} v^i(t, \mathbf{x}) = v^{i+1}(t, \mathbf{x})$ and $\lim_{t\downarrow t_{i-1}} u^i(t, \mathbf{x}) = u^{i-1}(t, \mathbf{x})$ for all $i \in [1:k]$, then $\mathbb{P}^{\text{mSBP}}$ of mSBP induced by following SDEs:*

$$d\mathbf{X}_t^\star = [f_t(\mathbf{X}_t^\star) + \sigma v^\star(t, \mathbf{X}_t^\star)]\, dt + \sigma d\mathbf{W}_t, \quad \mathbf{X}_0^\star \sim \rho_0. \tag{18a}$$

$$d\mathbf{Y}_t^\star = [-f_{T-t}(\mathbf{Y}_t^\star) + \sigma u^\star(t, \mathbf{Y}_t^\star)]\, dt + \sigma d\mathbf{W}_t, \quad \mathbf{Y}_0^\star \sim \rho_T, \tag{18b}$$

$$\text{where} \quad v^\star(t, \mathbf{x}) = \sum_{i=1}^k \mathbf{1}_{[t_{i-1}, t_i)}(t) v^i(t, \mathbf{x}), \quad u^\star(t, \mathbf{x}) = \sum_{i=1}^k \mathbf{1}_{(t_{i-1}, t_i]}(t) u^i(t, \mathbf{x}). \tag{18c}$$

Building upon Corollary 5, we introduce our Multi-Marginal Schrödinger Bridge Matching (MSBM) method to solve the mSBP. A cornerstone of MSBM is divide the global mSBP into local SBPs while maintaining the continuity of the composite drift functions $v^\star$ and $u^\star$ in (18c) across adjacent intervals, which guarantees a globally continuous diffusion process inducing $\mathbb{P}^{\text{mSBP}}$. Furthermore, by explicitly constraining each local SBs, $\mathbb{P}^i$, on its corresponding marginals $(\rho_{t_{i-1}}, \rho_{t_i})$, MSBM is designed to mitigate the accumulation of bias at intermediate marginals, as shown in Figure 1.

A key challenge of the MSBM is rigorously satisfying the continuity conditions at the boundaries of local controls: $\lim_{t\uparrow t_i} v^i(t, \mathbf{x}) = v^{i+1}(t, \mathbf{x})$ and $\lim_{t\downarrow t_{i-1}} u^i(t, \mathbf{x}) = u^{i-1}(t, \mathbf{x})$ for all $i \in [1:k]$. If these conditions are not met, discontinuities or "kinks" can arise at the intermediate time steps. Such kinks would imply that the overall path measure $\mathbb{P}^\star \neq \mathcal{M}^{\text{mm}}(\mathbb{P}^\star, \mathcal{T})$. This would, in turn, hinder the optimlaity for mSBP, because, following Proposition 3, the desired continuous Markov process is a fixed point of both $\mathcal{R}^{\text{mm}}$ and Markov projections $\mathcal{M}^{\text{mm}}$ under multiple time points $\mathcal{T}$:

$$\mathbb{P}^\star = \mathcal{R}^{\text{mm}}(\mathbb{P}^\star, \mathcal{T}) = \mathcal{M}^{\text{mm}}(\mathbb{P}^\star, \mathcal{T}). \tag{19}$$

To construct local SBs such that the continuity requirements for forming a valid global solution are met, thereby preventing the aforementioned kinks and ensuring (19), our MSBM introduces a shared global parametrization $v_\theta, u_\phi$ for its respective local controls $\{v^i, u^i\}_{i \in [1:k]}$ for each sub-interval, where each local controls are parallel updated with following aggregate objective function:

$$\tilde{\mathcal{L}}(\theta) = \sum_{i=1}^{k} \mathcal{L}(\theta, \mathcal{T}_i, \Pi_{\mathcal{T}_i}), \quad \tilde{\mathcal{L}}(\phi) = \sum_{i=1}^{k} \mathcal{L}(\phi, \mathcal{T}_i, \Pi_{\mathcal{T}_i}), \tag{20a}$$

where $\mathcal{T}_i = \{t_{i-1}, t_i\}$ define sub-intervals with local coupling $\Pi_{\mathcal{T}_i}$ for end-points marginals in interval $[t_{i-1}, t_i]$ and $\mathcal{L}$ is defined in (14) and (17) for forward and backward direction, respectively.

The MSBM training procedure, summarized in Algorithm 1, adapts the standard IMF algorithm presented in [45, Algorithm 1]. A key distinction in our MSBM approach is the parallel application of the IMF procedure to each local time interval, utilizing globally shared forward $v_\theta$ and backward $u_\phi$ across all local intervals. This parallel processing across sub-intervals contributes to a significant reduction in overall training time.

# 5 Related Work

The solution of SBP often utilize Iterative Proportional Fitting (IPF) [19], with modern adaptations learning SDE drifts for two-marginal settings [4, 9, 13, 49]. A distinct iterative approach, IMF, as featured in [37, 45], offers improved stability by alternating projections onto different classes of path measures. Moreover, emerging research also explores non-iterative algorithm [12, 38]. These methodologies primarily concentrate on the SB problem itself, iteratively refining path measures or directly computing the bridge measure. Moreover, the SB algorithm is studied under the assumption that the optimal coupling is given [27, 46]. While recent studies have extended foundational SB ideas to the multi-marginal setting of mSBP, research in this area remains relatively limited.

In multi-marginal setting, [8] extends the problem to phase space to encourage smoother trajectories and introduces a novel training methodology inspired by the Bregman iteration [5] to handle multiple marginal constraints. Relatedly, [44] presented an approach that, similar to our work, segments the problem across intervals; they learn piecewise SBs and use likelihood-based training to iteratively refine a global reference dynamic. While these methods are often IPF-based or focus on specific reference refinement strategies, our MSBM extends the previous IMF-type algorithm into multi-marginal setting and effectively handles multiple constraints. We demonstrate that our MSBM framework offers substantial gains in training efficiency. This enhanced efficiency is primarily attributed to its direct multi-marginal formulation that adeptly manages multiple constraints, thereby circumventing the computationally intensive iterative refinements common in IPF-based methods

Paralleling these SB-centric developments, other significant lines of work model dynamic trajectories by directly learning potential functions or velocity fields, often drawing from optimal transport or continuous normalizing flows. For instance, [18, 24–26] extend SBs to incorporate potentials or mean-field interactions, connecting to stochastic optimal control and earlier mean-field game frameworks [22, 41]. The broader field of trajectory inference from snapshot data, crucial for applications like scRNA-seq, has seen methods like [48] using CNFs with dynamic OT, and [15] employing Neural ODEs on learned data manifolds. More recently, [33, 34] offer variational objectives to learn dynamics from marginal samples.

# 6 Experiments

In this section, we empirically demonstrate the effectiveness of our MSBM. Specifically, our goal is to infer a dynamic model from datasets composed of samples from marginal distributions $\rho_t$ observed at discrete time points. We evaluate MSBM on both synthetic datasets and real-world single-cell RNA sequencing datasets, including human embryonic stem cells (hESC) [11] and embryoid body (EB) [32]. To ensure consistency and fair comparison, our experiments follow the respective experimental setups established by baseline methods. In particular, for the petal dataset, we adopt the experimental setup from DMSB [8], and for the hESC dataset, we follow SBIRR [44]. For the EB dataset, we perform evaluations on both 5-dim and 100-dim PCA-reduced data; here, we follow the 100-dim experimental setup of DMSB and the 5-dim setup from NLSB [18]. Accordingly,

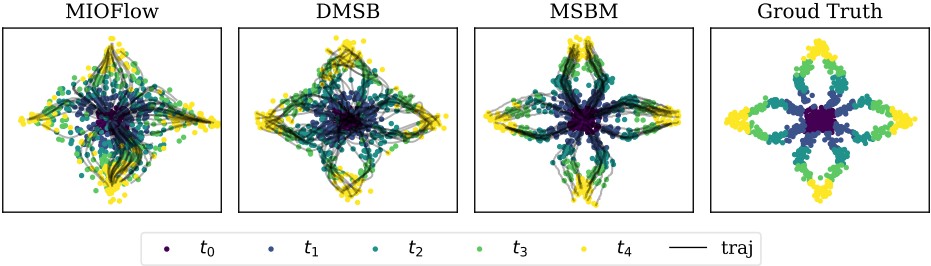

|  • $t_0$ | • $t_1$ | • $t_2$ | • $t_3$ | • $t_4$ | —— traj |

Figure 3: Comparison of generated population dynamics using MIOFlow, DMSB and MSBM on a 2-dim petal dataset. All trajectories are generated by simulating the dynamics from $\rho_{t_0}$.

276 we utilize evaluation metrics consistent with previous studies, including the Sliced-Wasserstein
277 Distance (SWD)[2], Maximum Mean Discrepancy (MMD)[14], as well as the 1-Wasserstein ($\mathcal{W}_1$)
278 and 2-Wasserstein ($\mathcal{W}_2$) distances. All experimental results reported are averaged mean value over
279 three independent runs with different random seeds. We highlight the best-performing results in **bold**
280 and the second-best results in blue. Further experimental details are provided in Appendix C.

## 6.1 Synthetic Data

**Petal** The petal dataset [15] serves as a simple yet complex challenge because it mimics the natural dynamics seen in processes such as cellular differentiation, which include phenomena like bifurcations and merges. We compare our MSBM with MIOFlow [15] and DMSB [8] in Figure 2. As shown in Figure 3, we observe that MSBM exhibits the most accurate and clearly defined trajectory, closely resembling the ground truth. Furthermore, Figure 2 demonstrates the evaluation results for the trajectories through $\mathcal{W}_2$ and MMD distances, highlighting that MSBM consistently outperforms MIOFlow and DMSB.

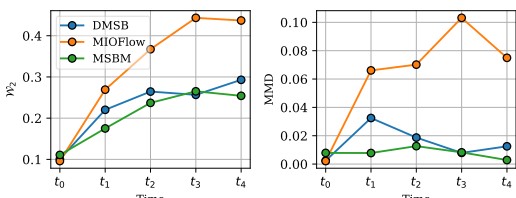

Figure 2: Evaluation results of $\mathcal{W}_2$ and MMD.

## 6.2 Single-cell Sequencing Data

294 We evaluated our MSBM on real-world single-cell RNA sequencing data from two sources: **1**) human
295 embryonic stem cells (hESCs) [11] undergoing differentiation into definitive endoderm over a 4-day
296 period, measured at 6 distinct time points ($t_0$:0 hours, $t_1$:12 hours, $t_2$:24 hours, $t_3$:36 hours, $t_4$:72
297 hours, and $t_5$:96 hours); **2**) embryoid body (EB) cells [32] differentiating into mesoderm, endoderm,
298 neuroectoderm, and neural crest over 27 days, with samples collected at 5 time windows ($t_0$:0-3 days,
299 $t_1$:6-9 days, $t_2$:12-15 days, $t_3$:18-21 days, and $t_4$:24-27 days). Following the experimental setup of
300 baselines, we preprocessed these datasets using the pipeline outlined in [48], and the collected cells
301 were projected into a lower-dimensional space using principal component analysis (PCA).

**hESC** To follow the experimental setup from SBIRR [44], we reduced the data to the first five principal components and excluded the final time point $t_6$ from our dataset, resulting in three training time points $\mathcal{T} = \{t_0, t_2, t_4\}$ and two intermediate test points $\mathcal{T}_{\texttt{test}} = \{t_1, t_3\}$. Our objective was to train the dynamics based on the available marginals at the training points in $\mathcal{T}$ and interpolate the intermediate test marginals at $\mathcal{T}_{\texttt{test}}$, which were not observed during training. Table 1 demonstrates that our proposed MSBM method performs competitively, achieving lower $\mathcal{W}_2$ distances.

Table 1: Performance on the 5-dim PCA of hESC dataset. $\mathcal{W}_2$ is compute between test $\rho_{t_i}$ and generated $\hat{\rho}_{t_i}$ by simulating the dynamics from test $\rho_{t_0}$.

| Methods | $\mathcal{W}_2 \downarrow$ | | Runtime |
|---|---|---|---|
| | $t_1$ | $t_3$ | hours |
| TrajectoryNet[†] | 1.30 | 1.93 | 10.19 |
| DMSB[†] | 1.10 | 1.51 | 15.54 |
| SBIRR[†] | **1.08** | 1.33 | 0.36 (0.38)* |
| **MSBM** (Ours) | 1.09 | **1.30** | **0.09** |

† result from [44].

**Embryoid Body** We validate our MSBM on both 5-dim and 100-dim PCA spaces. First, for the 5-dim experiment, we adopt the experimental setup from NLSB. Given 5 observation time points $\mathcal{T} = \{t_0, t_1, t_2, t_3, t_4\}$, we divide the data using train/test splits $\rho_{\mathcal{T}}^{\texttt{tr}}/\rho_{\mathcal{T}}^{\texttt{te}}$, with the goal of predicting population-level dynamics from $\rho_{t_0}^{\texttt{tr}}$. Similar to NLSB, we train the dynamics based on $\rho_{\mathcal{T}}^{\texttt{tr}}$ and

Table 3: Performance on the 100-dim PCA of EB dataset. MMD and SWD are computed between test $\rho_{t_i}^{\texttt{te}}$ and generated $\hat{\rho}_{t_i}$ by simulating the dynamics from test $\rho_{t_0}^{\texttt{te}}$.

| Methods | MMD ↓ | | | | SWD ↓ | | | |
|---|---|---|---|---|---|---|---|---|
| | Full | $t_1$ | $t_2$ | $t_3$ | Full | $t_1$ | $t_2$ | $t_3$ |
| NLSB[18] | 0.66 | 0.38 | 0.37 | 0.37 | 0.54 | 0.55 | 0.54 | 0.55 |
| MIOFlow[15] | 0.23 | 0.23 | 0.90 | 0.23 | 0.35 | 0.49 | 0.72 | 0.50 |
| DMSB[8] | 0.03 | 0.04 | 0.04 | 0.04 | 0.16 | 0.20 | 0.19 | 0.18 |
| **MSBM** | 0.02 | 0.04 | 0.04 | 0.05 | 0.11 | 0.18 | 0.17 | 0.19 |

† result from [8].

Figure 4: Comparison of generated population dynamics using DMSB and MSBM on a 100-dim PCA of EB dataset. The plot displays the first two principal components as the x and y axes, respectively.

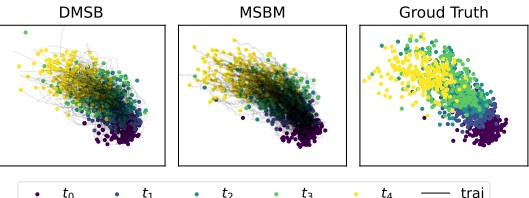

evaluate the $\mathcal{W}_1$ distance between $\rho_{t_i}^{\texttt{te}}$ and the generated $\hat{\rho}_{t_i}$ from previous test snapshot $\rho_{t_{i-1}}^{\texttt{te}}$. In Table 2, we find that MSBM outperforms several SB methods.

For the 100-dim experiment, we borrow the experimental setup from DMSB, where the goal is predict population dynamics given that observations are available for all time points $\mathcal{T}$ (denoted as Full in Table 3), or when one of the snapshot is left out (denoted as $t_i$ in Table 3, where snapshot $\rho_{t_i}^{\texttt{tr}}$ at $t_i$ is excluded during training). The high performance in this task represent the robustness of the model to accurately predict population dynamics. In Table 3, MSBM consistently yields performance improvements. Moreover, as shown in Figure 4, the trajectories and generated marginal distributions $\hat{\rho}_{\mathcal{T}}$ in PCA space further justifies the numerical result and highlights the variety and quality of the samples produced by MSBM.

**Computational Efficiency**   For an fair comparison of training efficiency against recent multi-marginal SB algorithms, we benchmarked DMSB and SBIRR on the identical hardware configuration employed for MSBM (denoted by * in Table 1). On the hESC dataset, MSBM achieved a runtime over 4× faster than SBIRR. Furthermore, on the petal and 100-dim PCA of EB dataset, MSBM significantly outperformed DSMB in training speed, with detailed results presented in Figure 5.

This enhanced computational efficiency primarily originates from core algorithmic differences. SBIRR, for example, utilizes maximum likelihood training, which requires extensive gradient computations and the storage of all intermediate paths. DMSB employs an IPF-type objective with Bregman Iteration [5]. In contrast, MSBM directly optimizes controls using an IMF-type objective, which not only eliminates the need to store intermediate states but also facilitates parallel computation across sub-intervals. This approach substantially promotes faster convergence of the algorithm.

Table 2: Performance on the 5-dim PCA of EB dataset. $\mathcal{W}_1$ is computed between test $\rho_{t_i}^{\texttt{te}}$ and generated $\hat{\rho}_{t_i}$ by simulating the dynamics from previous test $\rho_{t_{i-1}}^{\texttt{te}}$.

| Methods | $\mathcal{W}_1$ ↓ | | | | |
|---|---|---|---|---|---|
| | $t_1$ | $t_2$ | $t_3$ | $t_4$ | Mean |
| Neural SDE[21] | 0.69 | 0.91 | 0.85 | 0.81 | 0.82 |
| TrajectoryNet[48] | 0.73 | 1.06 | 0.90 | 1.01 | 0.93 |
| IPF (GP)[49] | 0.70 | 1.04 | 0.94 | 0.98 | 0.92 |
| IPF (NN)[4] | 0.73 | 0.89 | 0.84 | 0.83 | 0.82 |
| SB-FBSDE[9] | 0.56 | 0.80 | 1.00 | 1.00 | 0.84 |
| NLSB[18] | 0.68 | 0.84 | 0.81 | 0.79 | 0.78 |
| OT-CFM[47] | 0.78 | 0.76 | 0.77 | 0.75 | 0.77 |
| WLF-SB[34] | 0.63 | 0.79 | 0.77 | 0.75 | 0.73 |
| **MSBM (Ours)** | 0.64 | 0.73 | 0.72 | 0.73 | 0.71 |

† result from [18], ‡ result from [34].

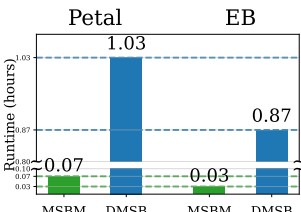

Figure 5: Training time

# 7   Conclusion and Limitation

This paper revisits previously established frameworks for the SBP, extending them to the mSBP. Specifically, we introduce a computationally efficient framework for mSBP, termed MSBM, which builds upon existing SBM methods [37, 45]. MSBM is tailored for various trajectory inference problems where snapshots of data are available at multi-marginal time steps. Through the successful adaptation of the IMF algorithm to this multi-marginal setting, our approach significantly accelerates training processes while ensuring accurate dynamic modeling when compared to existing methods.

Despite these advantages, the performance degradation of MSBM is more pronounced than that of DMSB when a time point is omitted in Table 3. This may occur because the including velocity term could better accommodate unknown trajectory. Furthermore, the current MSBM framework is restricted to the case involving snapshot data samples, highlighting a need for enhancements to address problems with continuous potentials, such mean-field games [18, 24–26].

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
