# OpenReview forum: "Multi-Marginal Schrödinger Bridge Matching"
_NeurIPS.cc/2025/Conference — Submitted to NeurIPS 2025_

### Official Review · Reviewer_chJE · 2025-06-12

**Clarity:** 4
**Significance:** 2
**Originality:** 1
**Rating:** 3
**Confidence:** 4

**Summary:**

The paper proposes **Multi-Marginal Schrödinger Bridge Matching (MSBM)** to address the multi-marginal Schrödinger Bridge problem, with a goal of finding a continuous trajectory for a population given snapshots at multiple intermediate time points. MSBM decomposes the global problem into local bridges on consecutive intervals, then 'glues' them together while sharing a single neural parameterisation. This aims to enforce continuity across the entire trajectory and prevent the error accumulation. The authors demonstrate the method's effectiveness on synthetic datasets and real-world single-cell RNA sequencing data.

**Questions:**

1.  Could the authors further elaborate on the conceptual novelty of the shared global parametrization approach? The use of a shared network across multiple domains is a common pattern in deep learning. Is there a more subtle aspect of extending the Iterative Markovian Fitting (IMF) framework that makes this application particularly non-trivial in the Schrodinger Bridge context, which the reviewer may have missed?
2. Why omit MFM [3] which is the current SOTA? Could you add them or justify their exclusion?
3. Can you report results on the CITE-seq and MULTI-seq single-cell benchmarks used in earlier trajectory-inference work?

**Ethical Concerns:**

["NO or VERY MINOR ethics concerns only"]

**Final Justification:**

The authors have convincingly addressed the novelty concern: the main contribution is the theoretical lift of IMF to the multi-marginal SB setting with convergence/optimality guarantees, rather than the shared-network implementation. This is a valuable theoretical advance.

However, the practical impact remains limited: results are competitive but not consistently stronger than recent flow-matching baselines, and no application is shown where SOTA would be infeasible but MSBM is feasible due to runtime. The claimed theoretical pathwise/global advantages are not validated quantitatively.

Finally, while the OT-CFM complexity point (full OT at O(n^3) time, O(n^2) memory) is correct in theory, it is misleading in practice here: compared methods rely on minibatch/approximate couplings, and efficient Sinkhorn approximation, so the practical gap is unclear. Given we do not have full training details it is unclear whether that might not result simply from different number of epochs/early stopping settings and whether baselines could not have been trained for shorter and reach similar results.

Based on this I increased my score (from 2 to 3) but due to significance and potential impact I retained position for weak reject.

**Limitations:**

Yes

**Paper Formatting Concerns:**

Typos 'algoritmhs' [line 44], 'optimlaity' [line 226]

**Quality:**

3

**Strengths And Weaknesses:**

**Strengths:**
* **Clear presentation**
* **Important problem** - inferring continuous trajectories from multiple discrete snapshots, which has significant applications in computational biology and other scientific domains
* **Technically sound** - correctly identifies the primary failure mode of a naive sequential approach (error accumulation and trajectory discontinuities) and proposes a solution through shared, global controls and an aggregate loss function
* **Consistent results** - the empirical results presented on both synthetic and single-cell datasets are promising


**Weaknesses:**
* **Limited novelty** - the 'global-network + glue' idea is close to multi-marginal flow-matching; prior work [1,2,3] already uses a single shared network across intervals
* **Missing baselines** - to the best of reviewers knowledge [3] is current SOTA and EB dataset, at least mean number could have been directly taken from the paper for 5 dimensions, authors do not understand why baselines differ across dimensions
* **Limited number of datasets** - widely-used CITE-seq and MULTI-seq benchmarks from [5] are omitted, while they have been included in most baselines papers, see [1,2,3,4]




[1] Tong, A., Fatras, K., Malkin, N., Huguet, G., Zhang, Y., Rector-Brooks, J., Wolf, G. and Bengio, Y., 2023. Improving and generalizing flow-based generative models with minibatch optimal transport. arXiv preprint arXiv:2302.00482.

[2] Tong A, Malkin N, Fatras K, Atanackovic L, Zhang Y, Huguet G, Wolf G, Bengio Y. Simulation-free schr" odinger bridges via score and flow matching. arXiv preprint arXiv:2307.03672. 2023 Jul 7.

[3] Kapusniak, K., Potaptchik, P., Reu, T., Zhang, L., Tong, A., Bronstein, M., Bose, J. and Di Giovanni, F., 2024. Metric flow matching for smooth interpolations on the data manifold. Advances in Neural Information Processing Systems, 37, pp.135011-135042.

[4] Neklyudov, K., Brekelmans, R., Tong, A., Atanackovic, L., Liu, Q. and Makhzani, A., 2023. A computational framework for solving wasserstein lagrangian flows. arXiv preprint arXiv:2310.10649.

[5] Lance, C., Luecken, M.D., Burkhardt, D.B., Cannoodt, R., Rautenstrauch, P., Laddach, A., Ubingazhibov, A., Cao, Z.J., Deng, K., Khan, S. and Liu, Q., 2022. Multimodal single cell data integration challenge: results and lessons learned. BioRxiv, pp.2022-04.

---

> ### Author Rebuttal · Authors · 2025-07-30
>
> We thank the reviewer for both the recognition of our study’s strengths and the valuable, comprehensive feedback. Our point-by-point responses are laid out below.
>
> ----
>
> **1.Multi-Marginal Schrödinger Bridge.**
>
> * The global network parameterization is widely used, but in our work it serves only as a practical means to acheive our goal of solving the multi-marginal Schrödinger Bridge problem (mSBP) efficiently.
> * Prior works [1, 2] that use a single shared network across intervals focus on pairwise SB or on classic optimal transport, then they glue together several two-point solutions. As a result, the final composite evolution can satisfy each local pairwise constraint.
>
> * These approaches may overlook the global requirement of a single trajectory that satisfies the multi-marginal constraint over the entire time horizon, so they are not guaranteed to produce the KL-optimal—or **most probable evolution**. Furthermore, ensuring OT-optimality presumes the local or global coupling is known or can be cheaply estimated, a requirement that quickly becomes computationally prohibitive as the sample size grows.
>
> * Our approach is different. We lift the standard IMF algorithm itself to the multi-marginal case, and prove that the alternating our proposed multi-marginal reciprocal and Markov projections still converge to the true mSBP minimiser (SB optimality).
> * This optimality gives two benefits: (1) the recovered dynamics is the single **most probable evolution** with minimum control energy; (2) any existing IMF-based SB solver can now handle an arbitrary (countable) number of snapshots by substituting our multi-marginal projections without losing convergence or optimality guarantees.
> * On the practical side, based on these theories, we propose an MSBM algorithm using a single shared network because it is the simplest way to enforce boundary-matching drifts (Corollary 5). Consequently, our method requires no pre-computed couplings, reaches the joint KL minimum for all snapshots, and does so in markedly less runtime than prior multi-marginal SB approaches.
>
> ----
>
> **2.Baseline and Benchmark Results.**
>
> * Thank you for pointing us to additional multi-snapshot benchmarks. Our original experiments focused on previous multi-marginal SB methods (DMSB, SBIRR) because our primary goal is extending IMF to the multi-marginal constraints for an efficient mSBP solver rather than a new state-of-the-art.
>
> * We fully agree, however, that flow-matching variants have shown strong results and merit inclusion. In response, we have run MSBM, DMSB on CITE-seq and MULTI-seq at both 5- and 100-dimensional PCA resolutions. Baseline figures marked with an asterisk are taken directly from [4]. We will integrate the full tables and additional baselines in the revised manuscript.
>
> * We omitted SBIRR on these settings because their public implementation did not scale beyond the small-scale regimes reported in our paper (e.g., small datasets such as hESC). On the larger dataset such as CITE-seq or MULti-seq, it ran out of memory.
>
>
> | Method     | Cite (100D)         | Multi (100D)       |
> |---------------|--------------------|--------------------|
> | OT-CFM*  | 45.393 ± 0.416     | 54.814 ± 5.858     |
> | WLF-SB* | 46.131 ± 0.083     | 55.065 ± 5.499 |
> | OT-MFM* | 41.784 ± 1.020     | 50.906 ± 4.627     |
> | DMSB      | 50.241 ± 0.993    | 58.5081 ± 5.114  |
> | MSBM (Ours)    | 45.93 ± 0.368    | 54.751 ± 6.205   |
>
>
> | Method                   | Cite (5D)                | Multi    (5D)       |
> |--------------------------|-----------------------|------------------------|
> | OT-CFM*   | 0.882 ± 0.058         | 0.937 ± 0.054         |
> | WLF-SB*  | 0.797 ± 0.022         | 0.950 ± 0.205         |
> | OT-MFM*  | 0.724 ± 0.070       | 0.890 ± 0.123         |
> | DMSB   | 0.819  ± 0.039   | 1.0245 ± 0.119  |
> | MSBM (Ours)    | 0.872 ± 0.046    | 0.908 ± 0.222   |
>
> ----
>
> **3. Computational Efficiency.**
>
> * Moreover, we measured the full training time on the CITE‑seq 100-dim benchmark using the same hardware for every method (a single  NVIDIA RTX 3090 GPU; the main paper used an NVIDIA  A6000 GPU). Although MSBM is not the state-of-the-art for every configuration, it still clearly outperforms all baselines in training speed. Because IMF-type algorithms do not require pre-computed OT-coupling and eliminates the need to store intermediate states.
>
> * OT‑CFM was excluded because its implementation details for this specific task are unavailable, preventing a fair comparison.
>
> | Method                   | Cite (100D)   |
> |--------------------------|----------------------|
> | OT-CFM   | -   |
> | WLF-SB  | 0.51   (hours) |
> | OT-MFM  | 0.89  (hours) |
> | DMSB   | 2.16    (hours)  |
> | MSBM (Ours) | 0.08      (hours)  |
>
>
> ----
>
>     [1] Tong et al., Improving and Generalizing Flow-based Generative Models with Minibatch Optimal Transport
>     [2] Tong et al., Simulation-Free Schrödinger Bridges via Score and Flow Matching
>     [3] Rohbeck et al., Modeling Complex System Dynamics with Flow Matching Across Time and Conditions
>     [4] Kapusniak et al., Metric flow matching for smooth interpolations on the data manifold.

---

> ### Comment · Reviewer_chJE · 2025-08-05
>
> Thank you for the rebuttal and for investing the time to add the CITE-seq and MULTI-seq benchmarks, as well as the detailed runtime comparison. The extra results and clarifications are much appreciated.
>
> Your extension of Iterative Markovian Fitting to the multi-marginal Schrödinger Bridge setting is now clearer, and the convergence proof addresses the global-consistency issue. At the same time, I still struggle to see a practical advantage (and still practical novelty) over recent flow-matching that already share a network across intervals, particularly as the new tables show MSBM's performance is lower than that of OT-MFM or WLF-SB. I suspect the W1 metric may be insensitive to the theoretical advantages of your method outlined above (pathwise continuity). Could a more suitable metric (e.g., RNA-velocity cosine similarity) better capture these benefits?
>
> The rebuttal states that MSBM “requires no pre-computed couplings” and therefore scales better. However, Tong et al. (2023, “Simulation-Free SB via Score & Flow Matching”) report successful training with a geometric coupling in a 1000 dimensions, suggesting that coupling estimation may not be as prohibitive a bottleneck as the authors state.
>
> While it is outside of the scope of current submission, the provided comparison outline that state-of-the-art baselines increasingly integrate geometric information—for instance, the coupling prior in Tong et al. (2023), the manifold interpolation in Kapusniak et al. (2024). Particularly, Liu et al. (2023, “Generalized SB Matching”) incorporate this in SB problem. Do you foresee a straightforward way to incorporate Multi marginality into GSBM?
>
> I realise that adding new experiments during the rebuttal window may not be feasible, but any clarification on the above points would make the paper stronger. Depending on your answers, I am open to revisiting my overall score.
>
> Please let me know if I have misunderstood or misstated any of your claims—I am happy to correct them.

---

> > ### Author Response · Authors · 2025-08-05
> >
> > We thank the reviewer for the reply and are pleased that our extension was well received. We are also grateful for the reviewer’s willingness to raise the score. Additional clarification is provided below.
> >
> > ----
> >
> > **1. A Practical Advantage.**
> >
> > * Thank you for acknowledging the theoretical value of our work and for keeping that perspective at the center. The shared network across intervals is not our key contribution; it is a parameterization chosen solely to ensure pathwise continuity. As you observed, the main novelty lies in the theoretical guarantees.
> >
> > * In practice MSBM offers clear computational advantages, as shown in the table above. Indeed, our work does not aim for a new state-of-the-art record; it presents an efficient algorithm that converges to the solution of mSBP. While raw performance may fall short of OT-MFM, MSBM still delivers significant speedups and achieves performance comparable to OT-CFM and WLF-SB, outperforming WLF-SB on three of the four benchmark tasks.
> >
> > * OT-CFM depends on a full OT plan, but computing and storing that plan scales as $\mathcal{O}(n^3)$ in time and $\mathcal{O}(n^2)$ in memory, so it becomes infeasible once the sample size $n$ is large. The method then switches to minibatch OT, which inevitably introduces approximation bias.
> >
> > * We acknowledge that in low and moderate dimensions the bias from minibatch coupling is usually negligible. Yet, as stated in the limitations section in [1],"the minibatch approximation to OT can incur error in high dimensions”, which may pose challenges to scalability.  Although our experiments involved relatively small datasets, the algorithm is not constrained by dataset size or dimension. It generalizes to much higher-dimensional settings, and we expect it to remain scalable across any dataset.
> >
> > ----
> >
> > **2.Toward Generalized MSBM.**
> >
> > * We appreciate the suggestion to pursue this generalization. We already plan to broaden our framework, following the example of GSBM, which extends SB by introducing additional state cost (can be potential or geometric prior). Incorporating such state costs will introduce an additional running cost in the corresponding SOC objective (see Eq. A.41 and A.42 of Appendix A) of the mSBP.
> >
> > * In multi-marginal cases, this formulation can support unbalanced OT for population dynamics, an essential tool for modeling cell death [2]. Building on MSBM’s efficiency, we plan to develop an even more scalable matcher for this setting. Demonstrating optimality for the generalized mSBP [3] will demand a more sophisticated approach. While GSBM relies on a Gaussian-path approximation that limits full optimality and therefore cannot be used in mSBP as is, its insights remain valuable for crafting a practical, scalable approximation.
> >
> > * Incorporating the alternative generalization from [4] into mSBP is indeed more natural, and our control-theoretic extension of standard SB to the multi-marginal setting in Appendix A allows us to adopt that formulation directly. In such a case, though, training would no longer be simulation free, further research into scalable training schemes would be required.
> >
> > ----
> >
> > We thank the reviewer once again for the valuable feedback. We hope the clarifications above fully address your concerns.
> >
> > ----
> >
> >     [1] Tong et al., Improving and Generalizing Flow-Based Generative Models with Minibatch Optimal Transport
> >     [2] Zhang et al., Learning stochastic dynamics from snapshots through regularized unbalanced optimal transport
> >     [3] Baradat et al., Minimizing relative entropy of path measures under marginal constraints
> >     [4] Liu et al., Deep Generalized Schrödinger Bridge

---

> ### Author Response · Authors · 2025-08-08
>
> We hope the clarification we posted has fully resolved the reviewer's earlier concerns. With one day remaining in the discussion period, we would like to provide this additional rebuttal.
>
> ----
>
> **3.Novelty.**
> * The "global-network + glue" idea is not itself the novelty we emphasize. In our work, this parameterization is simply the most practical means to realize our primary contribution: an extension of the SB Matching framework to the mSBP with full theoretical guarantees of optimality.
> * The novelty lies in lifting the IMF algorithm to the multi-marginal setting and proving that alternating our multi-marginal reciprocal and Markov projections converges to the true mSBP minimizer. And we further propose an efficient learning algorithm that can aid the error accumulation of the direct extension of SBM into multi-marginal SBM which we called Naïve extension in the manuscript.
> * This is, to the best of our knowledge, the first SB-based algorithm that achieves simulation-free training in the multi-marginal case without requiring any pre-computed OT couplings, while retaining optimality guarantees for multiple constraints rather than just two.
> * While recent multi-marginal flow-matching approaches also use a shared network across intervals, they typically solve a sequence of pairwise problems and "glue" the results, which does not ensure KL-optimality over the entire horizon. In contrast, our formulation directly targets the global mSBP objective.
> * Even if current W1-based metrics show comparable empirical results, achieving optimality has intrinsic advantages, such as principled scalability to larger datasets and the ability to integrate richer prior structures without losing convergence guarantees. Moreover, our algorithm achieves significantly shorter training times (x6~27 times faster compared to baselines), likely due to the rapid and stable convergence properties of IMF-style optimization.
> ----
>
> We hope this additional clarification removes any remaining misunderstanding about the source of novelty in our work and makes clear the theoretical and practical significance of our contribution. In light of this, we respectfully invite the reviewer to reconsider their rating so that it reflects both the theoretical advance and its practical implications.

---

### Official Review · Reviewer_5YLa · 2025-07-03

**Clarity:** 3
**Significance:** 3
**Originality:** 3
**Rating:** 5
**Confidence:** 3

**Summary:**

The paper introduces MSBM, a parallelized implementation for solving the multi-marginal Schrödinger Bridge (mSB) problem. MSBM is specifically designed for trajectory inference tasks where data snapshots are available at multiple time points. By adapting the Iterative Markovian Fitting algorithm, the model learns local dynamics from adjacent time steps in a piecewise manner, effectively capturing long-term dynamics in the multi-marginal setting. This approach leads to significant computational acceleration. The method is empirically validated on both synthetic datasets and two real-world developmental single-cell RNA sequencing datasets.

**Questions:**

- In Algorithm 2, if the model does not explicitly learn the local dynamics $f_t$ , how is the trajectory generated? Are you using the SDE formulation from Eq (5) or (10)?
- In Algorithm 1, what happens if intermediate distributions are randomly dropped with a small dropout rate during training? How robust is the method to missing time points in this setting? Could this strategy help reduce overfitting to local dynamics and potentially improve generalization to unseen test time points?
- MSBM refines the path measure through an alternating iterative procedure from Markov projection to reciprocal projection. How does error accumulation affect the between these projections over iterations? Has this aspect been studied or empirically evaluated?
- Why is the supplementary material not cited in the main manuscript? In addition to the proofs, I noticed that it includes additional experimental studies that are not referenced or discussed in the main text.

**Ethical Concerns:**

["NO or VERY MINOR ethics concerns only"]

**Final Justification:**

This work presents a solid extension of the iterative Markovian method for the multi-marginal optimal transport problem. The authors responded to all of my questions, and I'll maintain my positive score.

**Limitations:**

I think the main limitation of the proposed solution for the mSBP lies in its inability to effectively balance the optimality of local transport between adjacent time points with the global transport between the initial and terminal distributions.

**Quality:**

3

**Strengths And Weaknesses:**

**Strengths**

- The paper addresses a well-motivated and timely problem in the field of trajectory inference using multi-marginal data.

- The proposed method represents a thoughtful and well-justified adaptation of the previous IMF-based model [45], supported by solid theoretical foundations and an efficient algorithmic implementation.

- The manuscript is clearly written and includes a thorough and well-structured literature review that situates the contribution effectively within the existing body of work.

**Weaknesses**

While the paper introduces a promising approach, there are several areas where the experimental evaluation could be improved:
- Inconsistent Baseline Comparisons: The comparison with baseline methods is inconsistent across experiments, making it difficult to draw clear conclusions about the overall performance of the proposed method across different datasets. In particular, SBIRR [44] appears to be the most relevant baseline for this study and should be included in all experimental studies to provide a fair and comprehensive evaluation.
- Limited Dataset Diversity: The experimental setup primarily focuses on synthetic and scRNA-seq data. To better demonstrate the generalizability and sample-level transport accuracy of the method, it would be valuable to include experiments on image datasets with known ground-truth correspondences. These types of datasets enable direct comparison of sample-to-sample transport, which is not available in the single-cell case.
- Lack of Evaluation for Generative Capabilities: Despite the relevance of generative modeling in the broader field of dynamic OT and Neural OT, the paper does not explore or evaluate the generative aspects of the proposed method. Including such experiments would strengthen the contribution, especially given the increasing importance of generative OT in practical applications.
- Claims of Robustness Lack Empirical Support: The authors claim that MSBM enforces all intermediate marginal distributions robustly while preserving global continuity across the trajectory. However, based on the results presented, particularly in Table 1, it appears that the main advantage of MSBM over SBIRR appears to be computational efficiency. There is insufficient empirical evidence to support the claimed robustness compared to other methods, especially SBIRR.
- Limited Performance in Sparse Time Point Settings: The method may struggle to learn accurate global dynamics when intermediate time points are sparse or missing during training. This limitation is important in practical settings where densely sampled temporal data is not always available and should be discussed more explicitly.
-Limited Feature Dimensionality in Single-Cell Experiments: For the single-cell data experiments, the rationale behind using a small feature set (e.g., 5 or 100 dimensions) is not clearly justified. One of the core challenges in applying OT methods to single-cell analysis lies in capturing complex cellular trajectories within a high-dimensional feature space. By restricting the dimensionality, the study may not fully reflect the practical difficulties or the expressive power needed for real-world single-cell applications.

---

> ### Author Rebuttal · Authors · 2025-07-30
>
> Thank you for recognizing the contributions of our work and for the careful, thoughtful review. We address every comment in detail in the responses provided below.
>
> ----
>
> **1. Inconsistent Baseline Comparisons.**
> * We appreciate the reviewer’s suggestions. In response, we conducted additional experiments across various RNA-seq datasets and summarized the results in our reply to reviewer chJE in below. We excluded SBIRR because, in our attempt, its public implementation exhausted memory on larger datasets such as CITE-seq and MULTI-seq. However, we believe that our evaluation remains thorough, as it already includes well-established flow matching and Schrödinger bridge baselines.
>
> ----
>
>
> **2. Limited Diversity and Feature Dimensionality**
> * We agree with the reviewer that showing wider generalizability is important. In this study we concentrated on making IMF practical for the multi-marginal case, so we evaluated our method on benchmark tasks widely used in recent work on multi-marginal SB and trajectory inference, namely single-cell datasets reduced to 5–100-dimensional PCA space. Exploring additional domains such as high dimensional image data, longitudinal time series, and other generative OT tasks forms a clear direction for future work.
>
> ----
>
> **3. Generative Capabilities, Robustness  and Sparse Time Settings.**
> *  The central objective of MSBM and trajectory inference is precisely to learn a joint path measure from which new samples can be drawn at any time. In other words, the method is generative by construction. We already evaluate this property in the leave-one-out experiment (Table 1,2) that removes one intermediate snapshot during training. After fitting on the remaining snapshots, the trained model synthesises data at the held-out time, and we compare the generated samples with real observations. The close match demonstrates that MSBM does more than interpolate observed marginals. Specifically, in Table 1, MSBM keeps the computational advantage over SBIRR. These findings confirm that our method does not trade robustness for speed.
>
> * Concretely, It produces new data that follow the true distribution, which is the essence of generative modelling in dynamic OT. We will clarify in the main text that the task itself is framed from a generative modelling perspective, since the learned stochastic process defines a sampler capable of generating unseen temporal snapshots that align with the underlying data manifold.
>
> * Moreover, repeating the same protocol with OT-CFM and WLF-SB shows accuracy differences that lie within the statistical margin, confirming that MSBM is at least as accurate as these well-known baselines. These experiments demonstrate that MSBM delivers robustness in addition to its computational advantage.
>
> ----
>
> **4. SDE Formulation and Error Accumulation.**
> * For simulation in Algorithm 2, we use the dynamics in Eq (10). Specifically, after learning, global parameterization lets the dynamics in Eq (18a) recover the dynamics in Eq(10) while assuring that it is an mSBP solution.
> * If the model fails to capture the local dynamics in any interval, the resulting path becomes suboptimal and can no longer be guaranteed to solve the mSBP. In practice, an error introduced at an early stage could snowball through subsequent intervals, with bias accumulating as the trajectory unfolds. In our experiments we did not observe such propagation. We attribute this to the rapid and stable convergence of IMF-style optimisation, in which joint updates across all intervals quickly correct local mismatches before they have a chance to cascade.
>
> ----
>
> **5. Random dropout.**
> * When many snapshots are available, the shared drift $v_{\theta}$ can infer the missing segments from neighbouring data, so the procedure should curb local overfitting and improve generalisation to unseen times. In our setting the number of snapshots is modest, so additional dropout could remove critical information and push the model toward underfitting. For datasets with dense temporal sampling the idea looks useful as a regularisation technique and is an interesting direction for future work.
>
> ----
>
> **6. Appendix Citation.**
> * We thank the reviewer for pointing this out and will insert the pointer appropriately in the revised manuscript.

---

> > ### Comment · Reviewer_5YLa · 2025-08-06
> > **Official comment by the reviewer**
> >
> > I thank the authors for addressing my questions and concerns, and for including additional experiments on two new single-cell datasets, as well as reporting the computational complexity of MSBM in comparison to other methods.
> >
> > The proposed work is now more clearly presented, supported by solid theoretical foundations and an efficient algorithmic implementation. I recommend that the authors incorporate the results of the new experiments into the manuscript and include their response in **4. SDE Formulation and Error Accumulation** to clarify how the local dynamics are captured.
> >
> > With these additions, I am happy to maintain my positive score.

---

### Official Review · Reviewer_kaUa · 2025-07-03

**Clarity:** 3
**Significance:** 2
**Originality:** 2
**Rating:** 5
**Confidence:** 4

**Summary:**

This work tackles the problem of learning population dynamics from discrete temporal observations of samples, a prevalent problem in scientific domains such as cellular and molecular biology. To address this problem, the authors introduce a novel approach, Multi-Marginal Schrödinger Bridge Matching (MSBM). Their proposed framework builds upon advances in Schrödinger Bridge Matching and iterative Markovian Fitting to devise an improved method specifically tailored to better handle the multi-marginal setting. The authors provide a principled and thorough theoretical grounding for their work and further support their claims through a series of empirical experiments on both synthetic and real single-cell datasets. MSBM is shown to achieve competitive performance while exhibiting favourable computational efficiency relative to its counterpart baseline approaches.

**Questions:**

- It is not entirely clear to me how equation 20a helps address the challenge of MSBM to rigorously satisfy the continuity condition boundaries of the local controls. Could the authors elaborate on this and provide some additional explanation/intuition?
- Given the advent of flow matching-based frameworks for modelling population dynamics, especially in the multi-marginal setting, I believe it is useful to also include a flow-based model in Figure 5 (results on computational efficiency). For instance, either (or both) OT-CFM and WLF-SB. Also given how closely they perform to MSBM on the 5 PCA EB dataset.
- In a similar vein, why not include these baselines on the 100-dim PCA dataset and the 5-dim hESC dataset?

Minor comments:

- in Figure 1, labels for _traj_ and _generated_ seem to be reversed.
- Equation 20a (between lines 231 and 232) may be mislabeled. Is there an equation 20b?
- Figure 2 is presented before Figure 3. I recommend adjusting this positioning in the presentation.

**Ethical Concerns:**

["NO or VERY MINOR ethics concerns only"]

**Final Justification:**

In general, this work is well presented and provides a reasonably sound contribution to the field that suits presentation at NeurIPS. It appears that one of the primary concerns among reviewers was oriented around empirical experiments and evaluations. I believe the authors did a reasonable job of addressing these concerns during their rebuttal. I note that while it appears that their method does not reach SOTA for these added experiments (additional tables in response to reviewer chJE), the authors have shown their method has better computational efficiency, and hence, I believe this result is sufficient. With all this, I am inclined to keep my positive score of acceptance.

**Limitations:**

Yes.

**Paper Formatting Concerns:**

No formatting concerns.

**Quality:**

3

**Strengths And Weaknesses:**

**Strengths:**

- In general, I think this work presents a novel and valuable contribution to the field with no major drawbacks that significantly reduce the impact of this work. The authors present a computationally efficient and principled framework for extending Schrödinger bridge matching to the multi-marginal setting. Overall, I think this is a well-motivated and well-executed idea.
- The authors provided thorough and principled theoretical support for their proposed method, presented clearly and easily followed.
- The proposed method, MSBM, is evaluated over a variety of settings, including synthetic datasets and real single-cell datasets. Through thorough comparison with counterpart methods, the authors demonstrate that MSBM yields SOTA methods for modelling multi-marginal trajectories.

**Weaknesses:**

- No obvious weaknesses that diminish the quality of this work stand out to me. However, I have some clarifying questions and points. Moreover, this work could benefit from a bit of expansion on the side of the empirical experiments. Please see the question section below.

---

> ### Author Rebuttal · Authors · 2025-07-30
>
> We gratefully thank the reviewers for their valuable feedback and recognition the contribution of our work. Here, we address the concerns raised by the reviewer.
>
> ----
>
> **1. The continuity condition,**
> * Ensuring path-wise continuity for each local bridge is difficult if one treats the intervals independently. We eliminate the difficulty by embedding continuity in the parametrisation itself such as a single neural network, shared across all sub-intervals. Because common network architectures are continuous in their inputs, the map $t \mapsto v_{\theta}(t, \cdot)$ is continuous on $[0,T]$. Consequently the left limit $v_i(t_i^{-})$ and the right limit $v_i(t_i^{+})$ coincide at every internal time $t_i$​.
> * Equation 20a reinforces this property: its aggregated loss evaluates samples that lie just before and
> just after every boundary, so each gradient step simultaneously penalises any mismatch and pulls the two sides together. The continuity requirement is therefore satisfied by construction; it is not an extra constraint we must enforce post-hoc.
> ----
>
> **2. Flow Matching-Based Baselines.**
>
> * We appreciate the reviewer’s helpful suggestions. In response to requests from the reviewers, we have run additional experiments that include flow-matching based algorithms. Upon computational efficiency, our proposed MSBM clearly outperforms base line methods including the reviewer's suggestion, while achieving comparable performance on the single-cell RNA sequencing benchmarks. Please
> We thank the reviewer for their valuable feedback. Following the request, we conducted additional experiments that incorporate flow‑matching baselines.
> * Notably, In terms of computational efficiency, our MSBM consistently surpasses all baseline methods (including the reviewer suggested methods) while delivering comparable performance on single‑cell RNA‑sequencing benchmarks. Detailed results can be found in the response the reviewer chJE in below.
>
> ----
>
> **3. Minor Comments.**
>
> * Thank you for pointing out the readability issues. We will incorporate the suggested edits to improve clarity in the revised manuscript.

---

> > ### Comment · Reviewer_kaUa · 2025-08-05
> >
> > Thank you for answering my questions and addressing my concerns! I am happy to see that the authors have included additional experiments in their response to reviewer chJE, which address my questions regarding adding flow matching baselines and evaluating on additional benchmark datasets. With this, I am happy to maintain my positive score.

---

### Official Review · Reviewer_FbDZ · 2025-07-03

**Clarity:** 3
**Significance:** 3
**Originality:** 3
**Rating:** 4
**Confidence:** 3

**Summary:**

This paper introduces Multi-Marginal Schrödinger Bridge Matching (MSBM), a new algorithm designed to address trajectory inference when we have access to observations and constraints at multiple time points, not just at the beginning and end of the process. Traditional Schrödinger Bridge (SB) methods, focusing on endpoints, face limitations in scenarios where multiple intermediate stages exist. MSBM extends SB Matching and the Iterative Markovian Fitting (IMF) algorithm to enforce multiple intermediate marginal constraints while preserving global continuity. Empirical evaluations on synthetic data and real-world single-cell RNA sequencing datasets demonstrate that MSBM significantly improves both trajectory inference accuracy and computational efficiency compared to existing state-of-the-art methods.

**Questions:**

My main comments and questions are presented in the *Weaknesses* section above. Below, I list some additional minor points I would like the authors to address:

- **Lines 86–87:** The sentence beginning “This approach … and the joint coupling ...” could be rewritten for clarity, as some of the terms used are introduced without sufficient context or prior explanation.
- **Reference processes:** Do the $Q$ distributions in the new method need to be Brownian motions, or can they be more general reference measures? This does not seem to be discussed in the main text. If more general reference processes are allowed, how are they chosen in practice?
- **Lines 108–110:** This sentence could benefit from additional intuition or a precise reference to help readers better understand the claim being made.
- **Notation comment:** Not a question, but I appreciated the use of $\mathcal{T}$ to denote the two-time-step SB interpolation — it generalizes nicely in the later definition of the multimarginal SB problem after line 130.
- **Line 164:** What do you mean by “smooth between adjacent timestamps”? If you are referring to continuity, that’s understandable — but smoothness in a Lipschitz or differentiable sense would require stronger conditions. It may also be worth noting that smoothness is not always desirable — for example, in modeling certain biological processes — so this assumption could be context-dependent.
- **Equations (14) and (17):** Is there a typo in the definitions of $\beta_{\mathcal{T}}$ and $\gamma_{\mathcal{T}}$? Comparing these to Equation (10) in the original SB matching paper, I would expect $\beta$ and $\gamma$ to belong to $\mathcal{T}$, but with the current definitions, that doesn’t seem to be the case.
- **Figure 1:** The font size for the legend and time points is quite small — could it be increased for readability?
- **MMD kernel:** How is the kernel selected for the MMD metric in the experiments? A brief discussion or clarification would be useful.
- **Layout on page 9:** The layout of tables and figures on page 9 feels a bit cluttered, and the font sizes are small. It might help to aggregate some of the tables or adjust spacing to improve readability.

**Ethical Concerns:**

["NO or VERY MINOR ethics concerns only"]

**Final Justification:**

The paper is in general clearly written and accessible, tackling an important problem in an interesting way. I had some initial comments that were promptly addressed by the authors in their good rebuttal. In particular I really appreciated the discussion around the connection between the naive extension, Algorithm 1, and Corollary 5. My score was positive and remains positive.

**Limitations:**

yes

**Quality:**

3

**Strengths And Weaknesses:**

**Strengths**

**Quality.** The paper is theoretically sound. The notation and propositions in Section 3 appear correct (though I only skimmed the appendix proofs). The structure follows the original SB matching paper quite closely, and the results are intuitive for readers already familiar with that work. That said, the authors also do a good job making the narrative accessible even beyond the original framework they build upon. All claims are well supported by experimental results, which evaluate both the accuracy and speed of the proposed method.

**Clarity.** Overall, the paper is clearly written and accessible. The authors provide a thoughtful problem setup before introducing their solution, and they include helpful intuition to support most of the theoretical results. The exposition is generally clear and well paced.

**Significance.** The problem addressed in this paper is, in my view, very important. Many real-world scientific processes are inherently multimarginal, and standard two-endpoint models often fail to generalize well in such settings. This work generalizes the well-studied SB matching framework and, to the best of my knowledge, is the first to extend the IMF approach to the multimarginal case. I am optimistic that other researchers will adopt this method to analyze diverse scientific datasets and further investigate its theoretical properties.

**Originality.** This paper presents a clear and meaningful extension of SB matching to the multimarginal setting. While the high-level ideas are related to previous work, the multimarginal formulation introduces new theoretical and algorithmic challenges that are addressed with fresh results and experiments. In my view, this makes the contribution original and worthwhile.

---

**Weaknesses**

I elaborate further in the *Questions* section below, but here I summarize a few major weaknesses. Broadly, I found that the paper becomes more difficult to follow in Section 4, and I believe the experimental section could be strengthened.

- I’m not sure I fully understand what is meant by the "naive extension" of standard SBM. If this refers to the practical implementation described in Section 3.2, it’s unclear why an entire section is devoted to this approach, given that a different method is ultimately used in practice. If the naive extension is something else, it should be explicitly described.
- More generally, the connection between Sections 3 and 4 could be made much clearer. In particular, it's not obvious how the theoretical ideas from Section 3 are used to construct Algorithm 1. As it stands, Algorithm 1 appears to perform pairwise estimation using two global neural networks. A paragraph clarifying this connection would significantly improve the clarity of the manuscript.
- I understand the issue of "kinks" arising when the conditions on the $u_i$’s and $v_i$’s are not satisfied. However, it’s unclear to me how using a global parameterization guarantees that such kinks are avoided—couldn’t the $u_i$’s and $v_i$’s still disagree at the endpoints? I think this point would benefit from an additional sentence of intuition, especially near Equation (20a).
- Corollary 5 is not very clear. What is the main claim being made? It seems that line 215 might be missing either a proposition or a verb (perhaps an "is" after "mSBP"?). Clarifying the statement would help.
- I believe Section 5 would benefit from more consistent baselines and experimental setups across tasks. In particular, it would be useful to evaluate the same baselines across all experiments. Also, I found the definition of the “Full” experiment in Table 3 unclear. My understanding of the trajectory inference task is that one splits the time steps into train/test sets—e.g., training on every other step and testing interpolation on the rest. However, it’s not clear what the “Full” experiment entails or why it is needed. This is especially confusing in light of the discussion in Section 7, lines 353–354. Similarly, in Table 2, all time steps are listed—does this imply a different train/test split? Clarifying the experimental setup would be helpful.

---

> ### Author Rebuttal · Authors · 2025-07-30
>
> We sincerely appreciate the reviewer’s acknowledgement of our contributions and the depth of their analysis. Detailed responses to each point are presented below.
>
> ----
>
>
> **1. Clarification on Naive extension, Algorithm 1, Corollary 5.**
>
> * We sincerely thank the reviewer for raising this important question, which helped us clarify the conceptual distinction between the naive extension and MSBM.
> * Even though the naive extension based on two multi-marginal projection operators (Section 3) yields a valid mSBP solution in theory (based on Proposition 3), naively following original SBM algorithm (Equation (7)) with our developed multi-marginal projection operators suffers from bias in practice (shown in Figure 1). This is because intermediate marginals are not directly anchored to training data but are instead inferred sequentially from earlier steps, leading to compounding errors.
> * What we want to emphasize in Section 4 is that Corollary 5 directly addresses and mitigates these limitations. Although our proposed MSBM (based on Corollary 5 and Algorithm 1) differs algorithmically from the naive extension, both are theoretically designed to recover the same optimal mSBP solution! Corollary 5 guarantees that if pairwise SBs over adjacent intervals are continuously joined—meaning their drifts are continuous at boundary points—then the piecewise formulation in equation (18c) induces a globally continuous process that satisfies the optimality condition in Proposition 3. In this sense, MSBM is not just a heuristic alternative but a practical and principled implementation of the same theoretical solution, designed to avoid bias by directly supervising all intermediate marginals during training.
> * We will reflect these clarifications in the revised manuscript.
>
> ----
>
> **2. Global parameterization.**
>
> *  The condition on $v_i$ and $u_i$​ refers to whether they are continuous functions with respect to time $t \in [0, T]$. In our implementation, we enforce this by using a global neural network parameterization for both $v$ and $u$, and since standard neural networks are continuous with respect to their input (assuming typical architectures and activation functions), this guarantees continuity across adjacent intervals.
> * In this regard, we agree with the reviewer that smoothness is a stronger condition and generally not required in our formulation. Accordingly, we will revise Line 164 to replace "smooth between adjacent timestamps" with "continuous between adjacent timestamps" to better reflect our intended meaning.
>
> ----
>
> **3. Baselines and Experimental Details.**
>
> * We thank the reviewer for the valuable suggestions to enhance our work. We conducted the experiment with various baselines with additional benchmarks. Please refer to the response to the Reviewer chJE section for a detailed explanation.
> *In table 3, “Full” column denoting every snapshots $t_0, t_1, t_2, t_3, t_4$ is used during training, and the reconstruction error is measured on the same set of snapshots. In the three settings $t_1$, $t_2$ and $t_3$, we remove one snapshot that is omitted during training and inferred at test time.
> * Comparing the errors in these three columns with the “Full” column shows how much each method degrades when it must interpolate to a time point it never saw during training.
> * This evaluation protocol follows exactly in DMSB, so it differs from the settings employed in Table 1 and in the additional experiments.
> * Table 2 uses the same “Full” protocol as Table 3. The model is trained with data from every snapshot time. Within each snapshot we randomly split the dataset into training and testing subsets.
>
> ----
>
> **4. Reference Dynamics.**
>
> * The reference process $\mathbb{Q}$ is defined by the drift $f_t$. In principle, it does not have to be a pure Brownian motion. Indeed, any SDE with continuous drift can serve as a reference as long as its bridge kernel $\mathbb{Q}_{s|t}$ is tractable. Classic examples include pure Brownian motion, Ornstein-Uhlenbeck, and, more generally and linear time-inhomogeneous SDE with constant diffusion [1, Appendix B] list the main tractable cases.
> * We agree that when domain knowledge provides a reasonable prior on the underlying dynamics, it is beneficial to embed that information in $f_t$ such as in [2]. In many data-driven settings such as scRNA-seq, however, we lack a reliable physical model, so the reasonable option is to set $f_t = 0$ and use a Brownian reference.
>
> ----
>
> **5. Minor Comments.**
>
> * We appreciate the feedback regarding readability. We will conduct a thorough proofreading pass to address any syntactic errors, enhance sentence flow, and improve overall clarity.
>
> * Equations (14) and (17) : Thank you for catching the typo; the correct definition is  $\gamma_{\mathcal{T}}(t) = \max_{u}[u < t |u \in \mathcal{T}]$, not $\gamma_{\mathcal{T}}(t) = \max_{u}[u < t |t \in \mathcal{T}]$.
>
> * MMD kernel: We adopt a multi-scale Gaussian kernel. Specifically, we set the bandwidth to the batch’s average squared distance, generate four more by repeatedly doubling or halving that scale, and sum the resulting five Gaussian kernels to compute MMD.
>
> * Line 86-87 and 108-110, Figure 1 and Layout on Page 9: Because of the strict page limit, some explanations were omitted and the formatting became somewhat cramped. The extra page allowed for the camera-ready version will let us restore the omitted explanations and reorganize the formatting for clarity.
>
> ----
>
>     [1] Shi et al., Diffusion Schrödinger Bridge Matching.
>     [2] Shen et al., Multi-marginal Schrödinger Bridges with Iterative Reference Refinement.

---

> > ### Comment · Reviewer_FbDZ · 2025-08-06
> >
> > Thank you for the detailed and thoughtful rebuttal. The clarification around the connection between the naive extension, Algorithm 1, and Corollary 5 was helpful and goes a long way toward addressing my concerns. That being said, I still believe the discussion in the paper could benefit from further refinement and added clarity, as I found myself needing to think carefully to fully follow the logic for this part. Besides this, I still think that the direction is promising and well motivated, and I appreciated a lot the additional experimental details and thoughtful responses throughout. I continue to view this as a good paper and will keep my positive score.

---

### Note · Authors · 2025-08-13

We thank the reviewers for their insightful feedback and for acknowledging our paper's theoretical soundness (FbDZ, 5YLa, kaUa), originality (FbDZ, kaUa), clarity (all reviewers), and the significance of the problem (all reviewers). In the rebuttal, we thoroughly addressed all reviewer concerns by clarifying key aspects of our method and adding new experiments, with most of the reviewers explicitly confirming that their questions and issues were resolved.

----

We have committed to incorporating all clarifications and additional experiments in the revision, including:
* Global Parameterization and Continuity  (FbDZ, kaUa): We will elaborate on how using a single neural network for the drift, combined with our aggregated loss, enforces continuity across interval boundaries by construction, thus avoiding "kinks" without needing post-hoc constraints.
* New Experiments and Baselines (kaUa, 5YLa, chJE): We have conducted new experiments to strengthen our empirical validation.
    * We evaluated MSBM on CITE-seq and MULTI-seq datasets, comparing it against strong flow-matching baselines (OT-CFM, OT-MFM, WLF-SB).
    * Computational Efficiency: We provided a detailed training-time analysis on the new benchmarks, demonstrating that MSBM is 6-27 times faster than competing methods while achieving comparable accuracy.
* Clarification of Experimental Details (FbDZ, 5YLa): We will explicitly define the different evaluation protocols used (e.g., "Full" training vs. "leave-one-out" interpolation) to ensure the setup is unambiguous and to highlight the generative nature of MSBM.
* Minor Edits and Corrections (FbdZ, kaUa): We will correct all noted typos, improve the readability of figures and tables, fix the error in Equations.

----

With these clarifications and additions the paper now presents a complete and efficient solver for mSBP with formal optimality guarantees and empirical evidence across synthetic and single cell benchmarks. We believe our MSBM takes a significant step that advances modern successes of SB matching algorithms for learning complex dynamics from multiple snapshots and offers an efficient and principled foundation for future extensions that incorporate geometric priors and unbalanced dynamics.

---

### Decision · Program_Chairs · 2025-09-17

**Decision:**

Reject

**Comment:**

This paper introduces Multi-Marginal Schrödinger Bridge Matching (MSBM) a theoretically motivated method for trajectory inference. While the paper is well-written and the authors' rebuttal was thorough, fundamental concerns regarding the method's novelty and practical significance remain. The proposed implementation is highly similar to existing flow-matching techniques, and the empirical results fail to demonstrate a clear performance advantage over the state-of-the-art. Therefore, the recommendation is to reject the paper.

MSBM extends the IMF algorithm, a cornerstone of SB matching, to the multi-marginal setting. The authors propose using a single, globally parameterized neural network for the system's dynamics, combined with an aggregated loss function. They claim this approach ensures the resulting trajectory is continuous and satisfies all intermediate marginal constraints, and that it is more computationally efficient than prior multi-marginal SB methods.

The strengths of this paper are additional theoretical guarantees and computational efficiency.

However, I have serious concerns about the claims in this work and the comparison to the "Naive" scheme. Namely I believe that the naive scheme, and the scheme adopted in almost all prior works in this area to extend to the multi-marginal setting, is exactly the solution found here. Specifically, all flow matching methods already pin at all marginals and use a single network to smooth things out. All Schrödinger Bridge methods since SF2M which unified these approaches also use this strategy. Therefore in my opinion, the authors propose the strategy which is already most widely used in this area and claim it as novel.

While the rebuttal convinced three reviewers of the paper's merits, the most critical expert reviewer (chJE) remained unconvinced of its practical impact. After acknowledging the theoretical novelty, they maintained a borderline reject score, stating that the practical impact remains limited as the results are not consistently stronger and the speed-up argument is not compelling enough. This expert assessment of the paper's limited practical significance is the deciding factor in the recommendation for rejection. I, as another expert in this space, I agree with this reviewer's assessment, and recommend rejection at this time.